# A Python-Based Pipeline for Preprocessing LC–MS Data for Untargeted Metabolomics Workflows

**DOI:** 10.3390/metabo10100416

**Published:** 2020-10-16

**Authors:** Gabriel Riquelme, Nicolás Zabalegui, Pablo Marchi, Christina M. Jones, María Eugenia Monge

**Affiliations:** 1Centro de Investigaciones en Bionanociencias (CIBION), Consejo Nacional de Investigaciones Científicas y Técnicas (CONICET), Godoy Cruz 2390, Ciudad de Buenos Aires C1425FQD, Argentina; gabriel.riquelme@cibion.conicet.gov.ar (G.R.); nicolas.zabalegui@cibion.conicet.gov.ar (N.Z.); 2Departamento de Química Inorgánica Analítica y Química Física, Facultad de Ciencias Exactas y Naturales, Universidad de Buenos Aires, Ciudad Universitaria, Buenos Aires C1428EGA, Argentina; 3Facultad de Ingeniería, Universidad de Buenos Aires, Paseo Colón 850, Ciudad de Buenos Aires C1063ACV, Argentina; pmarchi@fi.uba.ar; 4Material Measurement Laboratory, National Institute of Standards and Technology, Gaithersburg, MD 20899-8392, USA; christina.jones@nist.gov

**Keywords:** data cleaning, preprocessing, Python, untargeted metabolomics, reference materials, data curation, system suitability, signal drift, quality control

## Abstract

Preprocessing data in a reproducible and robust way is one of the current challenges in untargeted metabolomics workflows. Data curation in liquid chromatography–mass spectrometry (LC–MS) involves the removal of biologically non-relevant features (retention time, *m/z* pairs) to retain only high-quality data for subsequent analysis and interpretation. The present work introduces TidyMS, a package for the Python programming language for preprocessing LC–MS data for quality control (QC) procedures in untargeted metabolomics workflows. It is a versatile strategy that can be customized or fit for purpose according to the specific metabolomics application. It allows performing quality control procedures to ensure accuracy and reliability in LC–MS measurements, and it allows preprocessing metabolomics data to obtain cleaned matrices for subsequent statistical analysis. The capabilities of the package are shown with pipelines for an LC–MS system suitability check, system conditioning, signal drift evaluation, and data curation. These applications were implemented to preprocess data corresponding to a new suite of candidate plasma reference materials developed by the National Institute of Standards and Technology (NIST; hypertriglyceridemic, diabetic, and African-American plasma pools) to be used in untargeted metabolomics studies in addition to NIST SRM 1950 Metabolites in Frozen Human Plasma. The package offers a rapid and reproducible workflow that can be used in an automated or semi-automated fashion, and it is an open and free tool available to all users.

## 1. Introduction

Recently, there has been an increasing awareness in the international metabolomics community about the need for implementing quality assurance (QA) and quality control (QC) processes to ensure data quality and reproducibility [1,2,3,4,5,6]. Challenges in untargeted metabolomics workflows are associated with pre-analytical, analytical, and post-analytical steps [1,2,3,4,5,7,8,9,10,11].

Liquid chromatography–mass spectrometry (LC–MS) is the analytical platform that provides the widest metabolome coverage in untargeted studies [7]. Several tools are broadly used in the metabolomics community for preprocessing LC–MS data, such as MZmine2, XCMS, MSDIAL, and workflow4metabolomics [8,12,13,14], among others. These software packages perform feature detection and correspondence, and provide an extracted data matrix as output for subsequent analysis.

Preprocessing LC–MS-based untargeted metabolomics data involves as well the removal of unwanted features (retention time, *m/z* pairs) to retain only those analytically robust enough for data analysis and interpretation [15,16]. To this end, several tools are available, such as SECIM-TOOLS [17] and nPYc-Toolbox [18]. Previous works have discussed strategies for data cleaning based on QC practices [2,19]. Different types of QC samples, i.e., system suitability samples, process blanks, pooled/intrastudy QC samples, process internal standards, and intralaboratory QC samples such as reference materials, allow the assessment of data quality within a study [6,19,20]. In particular, certified reference materials (CRMs) are commercially available standards that are used for accuracy control, method validation, and/or International System of Units traceability, while reference materials (RMs) (with non-certified values) are standards utilized for method harmonization and/or quality assessment [21]. RMs also allow data quality comparisons across different studies within a laboratory and across different laboratories and can, therefore, be used as interlaboratory QC samples [19,20,22]. In addition to data cleaning, data normalization strategies implemented before processing data and performing statistical analysis can dramatically influence results [23]. Different strategies for data normalization have been recently reviewed and classified as sample-based or data-based approaches [24].

Reproducibility of data curation steps contributes to improve data quality. Gathering all curation steps in one platform offers the advantages of semi-automatization, which can be translated into faster and more reproducible results. Python is a general-purpose, high-level programming language with several tools available for scientific computing [25,26]. Python packages include widely used machine learning frameworks, such as scikit-learn [27] and TensorFlow [28], allowing a seamless workflow from data curation to model generation. Moreover, libraries like Bokeh [29] make straightforward the construction of interactive web dashboards for data visualization and analysis. In addition, the pyopenms package [30] makes it possible to read MS data in the standard and open mzML format.

The present work introduces TidyMS, a package for the Python programming language for preprocessing LC–MS data for QC procedures in untargeted metabolomics workflows. The tool is open and free to all users to promote Findable, Accessible, Interoperable, and Reusable (FAIR) principles for the benefit of the community [31]. Two different applications are illustrated to show some of its capabilities, including a pipeline for an LC–MS system suitability check, analytical platform conditioning, and signal drift evaluation, as well as a pipeline for LC–MS data curation. Both applications were exemplified with the analysis of a new suite of candidate plasma reference materials for untargeted metabolomics (candidate RM 8231) currently under development by the National Institute of Standards and Technology (NIST; hypertriglyceridemic, diabetic, and African-American plasma pools), in addition to NIST SRM 1950 Metabolites in Frozen Human Plasma. These examples show that the package offers rapid and reproducible data preprocessing capabilities that can be used in an automated or semi-automated fashion and can be customized or fit for purpose according to the specific metabolomics application. These characteristics provide comparative advantages toward manual preprocessing actions. Source code, documentation, and installation instructions are available online at Github: https://github.com/griquelme/tidyms.

## 2. Results and Discussion

### 2.1. Package Implementation

TidyMS was designed with the goal of preprocessing and curating data from any untargeted LC–MS-based metabolomics study. After applying feature detection (e.g., peak picking and integration on chromatographic data) and feature correspondence (feature matching across different samples, e.g., chromatogram alignment) algorithms on the raw experimental data, an extracted data matrix of dimension N × K, where N is the total number of samples and K the total number of detected features, is obtained. The value of each (n, k) element of the extracted data matrix is associated with the chromatographic peak area of the k-th feature in the n-th sample. Different descriptors are associated with the data matrix, including the experimental exact mass and retention time (Rt) values for features, and run order for samples. The data curation process allows retaining analytically robust features that may be potentially identifiable. Several metrics are typically adopted by the metabolomics community to retain analytically robust features [19]. As well, minimum requirements should be fulfilled to pursue feature identification. These include the detection of the feature isotopic distribution, [32] with a monoisotopic peak exhibiting a large enough intensity value in the raw data to allow subsequent MS^n^ experiments. Even if there are several software alternatives available to perform data curation, to the best of our knowledge, none has been so far able to fully cover our proposed strategy. Specifically, it concerns the last two steps that require going back and forth between the data matrix and the raw data. In this context, the implemented package can deal with both a data matrix in tabular form and raw MS data. The TidyMS source code [33] and Jupyter notebooks with the results from Applications 1 and 2 [34] are freely available at Github.

#### 2.1.1. Raw Data Analysis Tools

Raw data analysis is mostly done through the MSData object (see Materials and Methods), which has methods for chromatogram creation and MS spectrum accumulation. Analysis of chromatographic and spectrum peaks are done using a continuous wavelet transform (CWT)-based peak picking algorithm [35]. Peak picking can be used with feature information from a data matrix to detect features in raw data, using the centWave algorithm [36] (see Application 1).

#### 2.1.2. Data Matrix Curation Tools

Methods for filtering features in the data matrix are encapsulated in the DataContainer, Filter, Corrector, and Pipeline objects (Figure 1). The DataContainer uses Pandas [26] at its core, and stores the data matrix and the descriptors associated with features and samples. The sample mapping attribute of the DataContainer object associates sample types to sample classes, and it is used to set a default behavior for the different filters applied to the data matrix. Available sample types are study samples, pooled/intrastudy QC samples, and blanks.

The Corrector and Filter objects manage the different correcting and filtering steps, respectively. For a correction step such as the blank correction, a subset of samples is used to generate a corrective action that is subsequently applied to the samples of interest. For filters, a metric is evaluated, and features are removed based on predefined bounds or an accepted range. To apply a correction or a filter, the user must specify not only the filter parameters but also the sample classes needed to generate the correction (or metric in the case of filters), as well as the sample classes in which the correction should be applied. To simplify the user input, each Processor object determines which subsets of sample classes to use with the sample mapping attribute of the DataContainer (see Materials and Methods). Default parameters for the filtering process were implemented based on the guidelines proposed by Broadhurst et al. [19]. Available Correctors in the package include the BlankCorrector that aims to remove the contribution of a feature present in blank samples. These are typically used to identify contaminants present in solvents and those potentially introduced by sample storage and handling. Several modes are available to estimate the blank contribution to a feature intensity. In addition, feature abundance can be corrected for signal drift and inter-batch effects. A pooled/intrastudy QC sample-based robust locally estimated scatterplot smoothing (LOESS) signal correction method introduced by Dunn and collaborators [37] is also included in the package with the BatchCorrector object. The latter removes from feature intensity the bias associated with time-related effects, such as signal drift. The interpolator used to generate the correction should be selected together with the fraction of samples to be used by the LOESS approach. By default, this fraction is optimized for each feature using a leave-one-out cross-validation (LOOCV) method (see Appendix A (Appendix A) for a detailed description of the considerations that should be taken before applying a batch correction on a data set).

Several filters are also included in the package. The PrevalenceFilter is based on the Detection Rate (DR) of a feature, i.e., the fraction of samples in which a feature is detected. The PrevalenceFilter evaluates the DR in study sample classes, ensuring that a feature is consistently detected in at least one sample class. Lower and upper bounds can be set for the filter as well as the mode in which the DR is computed. In an intraclass mode, the DR is computed for each class and the minimum value is used, whereas in a global mode the DR is computed using all samples from the selected classes. The VariationFilter works in a similar way as the PrevalenceFilter, but it uses the coefficient of variation or the relative median absolute deviation as a measure of sample dispersion. By default, pooled/intrastudy QC samples are used to remove features that are not measured in a robust way. The last filtering option available in the package is the DRatioFilter, which has also been described by Dunn and collaborators [19]. This filter uses the dispersion of each feature in pooled/intrastudy QC samples and in study samples to compute a ratio that addresses technical and biological variation. Features with high DRatio values are removed. The user can set a maximum value for an acceptable DRatio.

### 2.2. Application 1: System Suitability Check and Signal Drift Evaluation

The ability of the package to process raw data is illustrated for an LC–MS system suitability check. Figure 2 shows the run order of an untargeted metabolomics study starting with a zero injection, followed by a reconstitution solvent injection, and a subsequent process blank injection. This first set of injections contributed to reveal the presence of potential impurities or contaminants in the LC–MS system, and those potentially introduced along the sample preparation protocol. Next, a system suitability QC sample (SSS) consisting of five chemical standards was injected to assess *m/z*, retention time, and peak area, by comparison to their average values. Figure 3 illustrates results from ten consecutive technical replicates of the SSS to establish the mean and dispersion values for *m/z*, retention time, and peak area for each compound in the sample. These values were computed through the implementation of the centWave algorithm using centroid raw data as input, and a cluster-based feature correspondence algorithm (see Appendix A (Appendix A)) and were used to build an acceptance criteria of the LC–MS system performance for each authentic chemical standard. There is no unique way to define acceptance criteria for untargeted metabolomics, and each laboratory is encouraged to build their own. QA and QC common practices as well as criteria adopted by different laboratories working in untargeted metabolomics worldwide have been recently addressed by Evans et al. [6]. In this example, the acceptance interval for both *m/z* and peak area was defined as 2-fold the standard deviation of the mean, and three seconds for the retention time. Figure 3 shows the system suitability application by jointly plotting the values obtained for the SSS measured before and after the analysis of biological samples. The extracted ion chromatograms can also be visualized with the package, and are illustrated in Appendix A (Appendix A) for three chemical standards in the SSS. This QC procedure was also performed on pooled/intrastudy QC samples that were spiked with these same chemical standards as well as with L-Leucine-1-^13^C used as an internal standard (Appendix A (Appendix A)). This capability of the pipeline can be used to follow the instrument performance during a batch analysis.

After the system suitability sample was analyzed, eight injections of a pooled/intrastudy QC sample, addressed as “system conditioning QCs”, were required to equilibrate the analytical platform. Three additional consecutive injections of pooled/intrastudy QC samples were analyzed before running the study samples, composed of NIST candidate RM 8231 and NIST SRM 1950. These samples were analyzed in six blocks of four runs, each block being individually balanced, randomized, and bracketed by pooled/intrastudy QC samples. At the end of the batch, three additional technical replicates of pooled/intrastudy QC samples were injected, followed by a blank and an SSS with the goal of evaluating potential cumulative carryover as well as the analytical platform final status, respectively. As time-related systematic variation of metabolite responses is typically observed in untargeted data sets, the pipeline allows (i) building principal component analysis (PCA) models for QC assessment (Figure 4, Appendix A (Appendix A)), and (ii) applying a pooled/intrastudy QC sample-based LOESS batch correction [37]. Indeed, the pooled/intrastudy QC sample template illustrated in the sample list (Figure 2), with replicates at the beginning and the end of the batch, was designed to minimize interpolation issues associated with the LOESS correction process (see Appendix A). PCA score plots illustrated in Figure 4 (upper panel) and Appendix A show the presence of signal drift using both the whole data set and the pooled/intrastudy QC samples, respectively. The differences observed in the score plots shown in Figure 4 and Appendix A can be explained, considering that the biological variation is expected to be larger than the instrumental drift. In this regard, when the study samples and the pooled/intrastudy QC samples are used to build the PCA model (Figure 4), the temporal drift can be associated to PC2, but when only the pooled/intrastudy QC samples are used to build the model (Appendix A), the main source of variation is the signal drift, and can be observed along PC1. As expected, this trend disappears after data curation, as random error is the only source of variation (Appendix A). Based on the internal standard results displayed in Appendix A, the contribution of the sample preparation variation was negligible compared to the total variation and was, therefore, not included in this discussion.

### 2.3. Application 2: Analysis of Candidate Reference Standard Materials

The scheme illustrated in Figure 4 shows the different curation steps that were performed with the in-house Python-based data preprocessing pipeline for analyzing the NIST candidate RM 8231 and SRM 1950. The first step was implemented to discard features with retention time values lower than 90 s, as the system dead time was approximately 0.8 min. The second step allowed discarding features that were not present in pooled/intrastudy QC samples (see Appendix A) as a condition for performing the LOESS batch correction, suggested by results displayed in the upper PCA score plot. Subsequently, if a feature had a peak area in a plasma sample that was 10-fold or less than the maximum peak area in the solvent, zero volume injection, and process blanks of the same feature, then its peak area was set to 0. Otherwise, the mean peak area in those blanks was subtracted from the feature peak areas in the plasma samples. In this way, potential contaminants and signals from the solvent would be removed for further analysis. A subsequent filter based on the Relative Standard Deviation (RSD) of each feature in pooled/intrastudy QC samples was applied, retaining 1849 features in the data matrix. That is, all features with an RSD larger than 20% in pooled/intrastudy QC sample injections were eliminated. The Median Absolute Deviation (MAD) and the median were used to obtain a robust and unbiased estimation of the RSD, assuming a normal distribution and multiplying by the correspondent scaling factor. A 100% intraclass prevalence filter was subsequently applied with a threshold area value of 5, i.e., all area values below this threshold were set to zero. The prevalence filter was followed by the D-ratio filter [19], estimated as the MAD of pooled/intrastudy QC samples relative to the MAD of all study samples. This filter was used to remove features with zero or low biological information, setting as acceptance criterion D-ratio values lower than 10%, and leading to a final curated data matrix composed of 665 metabolic features. Data were finally normalized at the end of the pipeline by using the total area. Each step of the data curation process can be visually inspected by means of PCA models, as exemplified in Figure 4 with the score plots displayed for the raw and curated data sets. In addition, metrics such as RSD or D-ratio can be computed at each step of the curation pipeline to evaluate the characteristics of the different features.

### 2.4. Running Times

It is not our goal to compare the performance of TidyMS against available alternatives, but to show the reader and potential users that results can be obtained in a time scale of minutes. The results presented in Applications 1 and 2 were generated using a computer with an Intel 8 Core 1.60 GHz 8th generation i5 processor and 8 GB memory running Ubuntu Linux 18.04. The running time needed to analyze the 31 raw data files used to generate the results shown in Application 1 was 66 s. Most of the computing time was spent on feature detection (about 2 s per sample), but it has to be noted that only a subset of known *m/z* values associated with the five chemical standards was searched in the data. An untargeted analysis of each sample has an average running time of 45 s, this time being associated with the sensitivity used to generate the centroid data. Changing the signal-to-noise ratio (SNR) in this step can considerably reduce this time. The running time needed to complete the data curation pipeline in Application 2 was 9 min. Most of this time was spent on the batch correction step, which requires computing LOESS combined with LOOCV for each feature. The user should be aware that this time can be increased depending on the number of features and analytical batches in the data set.

### 2.5. Comparable Software Alternatives

As stated in the previous sections, the main objective of the present work was to design a tool to achieve data curation and quality assessment of metabolic features extracted in untargeted LC–MS-based metabolomics studies, in a fast, reproducible, and user-friendly way. Since Python is increasingly used in the scientific community, we aimed at bridging data generated in metabolomics workflows with state-of-the-art data analysis and machine learning tools that are currently available in this programming language (e.g., Pandas, scikit-learn, statsmodels). To the best of our knowledge, there are currently no other available tools in the Python programming language for preprocessing raw LC–MS data at a high level, although libraries like pyopenms make it possible to read raw data formats like mzML. Table 1 compares TidyMS with other open-source tools based on selected features. Since TidyMS is locally executed, we excluded cloud-based alternatives for comparison. XCMS online and Workflow4Metabolomics, which are online platforms widely used by the metabolomics community, are capable of fully covering the preprocessing workflow, making use of some of the packages described in Table 1. Despite the overlap in the data curation functionality of most software alternatives included in Table 1, TidyMS is mainly focused on providing specific tools to evaluate and assess data quality through the use of pooled/intrastudy QC samples. In this sense, there is a certain degree of overlap between TidyMS and both nPYc-Toolbox and Notame, which were made publicly available while we were developing TidyMS, highlighting the necessity in the field of tools to achieve reproducible data curation workflows. All in all, TidyMS spans from preprocessing raw data to curating a data matrix, and it is a flexible and versatile platform that allows straightforward extensions of the current pipeline based on new emerging strategies adopted by the metabolomics community.

### 2.6. Limitations and Perspectives

Since our feature correspondence algorithm was not thoroughly tested on untargeted data sets, it is not yet recommended to use the package as a full pipeline analysis of raw untargeted collected data. Instead, we suggest that users extract feature matrices with more mature software, such as XCMS or MZMine2, and use TidyMS for data curation and QC assessment in LC–MS untargeted metabolomics workflows. We plan to test and refine the correspondence algorithm in the following release of the package. In addition, we aim to extend its functionality to include tools for reducing the error rate of data curation problems associated with isotope and adduct clustering. As well, the package is in active development to correct bugs and to provide a better interface based on the user feedback.

## 3. Conclusions

TidyMS provides functionality to perform rapid analysis of raw data and data curation on untargeted metabolomics data sets and can be adapted to fit for different objectives. It also supports interactive visualization of data sets, which is needed to verify data quality throughout an experimental workflow. The implementation in Python allows easy integration of results into state-of-the-art tools for model generation, such as scikit-learn. By means of two different applications we demonstrated the capabilities of the package with pipelines for an LC–MS system suitability check, system conditioning for conducting untargeted metabolomics studies, signal drift evaluation, and data curation. The pipeline was successfully implemented to preprocess data corresponding to NIST candidate RM 8231 and SRM 1950 for metabolomics studies. Overall, we showed that the package offers a rapid and reproducible workflow that can be used in an automated or semi-automated way, and it is an open and free tool available to all users.

## 4. Materials and Methods

### 4.1. Chemicals

LC–MS grade acetonitrile, methanol, and acetic acid, purchased from Fisher Chemical (Raleigh, NC, USA) and ultrapure water with 18.2 MΩ·cm resistivity (Thermo Scientific Barnstead MicroPure UV ultrapure water system, Sunnyvale, CA, USA) were used to prepare solutions and chromatographic mobile phases. Leucine enkephalin was purchased from Waters Corp. (Milford, MA, USA). L-Leucine-1-13C (purity = 99%) was purchased from Eurisotop, Saint-Aubin, France). Lysophosphatidylcholine USP standard (approximate content of LysoPC 0:0/16:0 = 65%, approximate content of LysoPC 0:0/18:0 = 25%), L-phenylalanyl-L-phenylalanine and L-tryptophan (≥98%), were purchased from Sigma-Aldrich (St. Louis, MO, USA). Alogliptin was purchased from DC Chemicals (Shanghai, China).

### 4.2. Plasma Samples

SRM 1950 Metabolites in Frozen Human Plasma and candidate RM 8231 Frozen Human Plasma Suite for Metabolomics (hypertriglyceridemic, diabetic, and African-American plasma samples) were provided by the Metabolomics Quality Assurance and Quality Control Program led by NIST. All NIST plasma samples were collected after informed consent under approved Institutional Review Board (IRB) protocols reviewed by the NIST Human Subjects Protection Office. Details on the preparation of SRM 1950 have been published. Bioreclamation, Inc. (Hicksville, NY, USA) collected whole blood from 100 donors (1:1 male-to-female ratio) between 40 and 50 years of age after an overnight fast and a 72 h abstention from medication. Lithium heparin was used as the anticoagulant. The collected blood was centrifuged at 8000 × *g_n_* for 25 min at 4 °C to obtain plasma. Each donor’s plasma was thawed once. Subsequently, all donor plasma was blended together and aliquoted into individual vials to produce the SRM. The diabetic plasma material (RM 8231-1) was created by Solomon Park Research Laboratories, Inc. (Kirkland, WA, USA). It was created from a pool of 6 female donors and 5 male donors between 34 and 68 years of age after an overnight fast, each meeting the specified ranges for glucose (>126 mg dL^−1^) and triglycerides (<150 mg dL^−1^). Solomon Park Research Laboratories, Inc. also created the hypertriglyceridemic plasma material (RM 8231-2). The donor pool consisted of 11 male donors between 31 and 72 years of age after an overnight fast, each meeting the NIST-specified ranges for glucose (<100 mg dL^−1^) and triglycerides (>300 mg dL^−1^). The diabetic and hypertriglyceridemic plasma were obtained by centrifuging the respective collected blood at 251 rad s^−1^ for 10 min, followed by additional centrifugation at 397 rad s^−1^. Lithium heparin was used as the anticoagulant for both, and plasma from each donor was thawed once and blended to make the respective donor pools before aliquoting into individual vials. BioreclamationIVT (Westbury, NY, USA) collected whole blood from 16 donors (1:1 male-to-female ratio) between 20 and 25 years of age after an overnight fast using K2 ethylenediaminetetraacetic acid (EDTA) as the anticoagulant for the African-American plasma (RM 8231-3). The 16 units of blood were subsequently thawed and pooled by Solomon Park Research Laboratories similar to the diabetic and hypertriglyceridemic plasma materials. Samples were shipped on dry ice to Centro de Investigaciones en Bionanociencias, Consejo Nacional de Investigaciones Científicas y Técnicas (CIBION-CONICET, Ciudad de Buenos Aires, Argentina) and stored at −80 °C until use.

### 4.3. Sample Preparation

Plasma samples were thawed on a water–ice bath prior to sample preparation. Each plasma sample was split into 6 aliquots for individual sample preparation and stored at −80 °C. As well, a pooled/intrastudy QC was created from equal aliquots of each plasma sample and subsequently aliquoted and stored at −80 °C. After thawing samples in a water–ice bath, a 10 µL portion of 40 µmol L^−1^ isotopically labeled leucine solution was added to 50 µL of each plasma sample and QC sample to be used as internal standard. Subsequently, samples were vortex-mixed and 180 µL of methanol/acetonitrile (50:50 *v*/*v*) were added to precipitate proteins. Samples were vortex-mixed for 30 s, stored at −20 °C for 1 h, and centrifuged at 21,382 × *g_n_* for 20 min at 4 °C. After centrifugation, 180 μL of supernatant were transferred to new microcentrifuge tubes and frozen at −80 °C for 30 min after the addition of 180 μL of ultrapure water. Subsequently, samples were lyophilized for 38 h at −80 °C and 6.67 Pa using a Telstar LYOQuest-85 freeze-dryer (Telstar, Madrid, Spain). Subsequently, sample residues were thawed on a water–ice bath for 30 min, reconstituted in 150 μL of water/methanol (80:20 *v*/*v*), vortex-mixed for 30 s, centrifuged for 20 min at 21,382 × *g_n_* and 4 °C, and 140 μL of supernatant were transferred to LC vials and analyzed by ultra-performance liquid chromatography coupled to quadrupole-time-of-flight mass spectrometry (UPLC–QTOF–MS). For pooled/intrastudy QC samples, a 138.6 μL portion was transferred to LC vials and spiked with 1.4 μL of a system suitability QC sample (SSS), consisting of a chemical standard mixture. Process blank samples, consisting of ultrapure water, underwent the same process as plasma samples and were pooled in a single vial. The standard mixture solution (SSS) was prepared in ultrapure water containing the following compounds: leucine enkephalin 25 µmol L^−1^, lysophosphatidylcholine (0:0/18:0) 17.9 µmol L^−1^, L-phenylalanyl-L-phenylalanine 100 µmol L^−1^, L-tryptophan 100 µmol L^−1^, and alogliptin 50 µmol L^−1^. This solution (SSS) was diluted 100-fold to be used as a system suitability QC sample and analyzed before conditioning the analytical platform and at the end of the batch (Figure 2).

### 4.4. Ultra-Performance Liquid Chromatography−Mass Spectrometry (UPLC–MS)

UPLC–MS experiments were performed with a Waters ACQUITY UPLC I Class system fitted with a Waters ACQUITY UPLC BEH C18 column (2.1 × 100 mm, 1.7 μm particle size, Waters Corporation, Milford, MA, USA), coupled to a Xevo G2S QTOF mass spectrometer (Waters Corporation, Manchester, UK) with an electrospray ionization (ESI) source. The mass spectrometer was operated in positive ion mode. The typical resolving power and mass accuracy of the Xevo G2S QTOF mass spectrometer were 27,102 FWHM and 2.7 ppm at *m*/*z* 556.2771, respectively. Gradient elution was used in the chromatographic separation method. Mobile phases consisted of water with 0.1% acetic acid (mobile phase A) and acetonitrile (mobile phase B), and the method used the following gradient program: 0−1 min 10% B; 1−2.5 min 10−15% B; 2.5−4 min 15−22% B; 4−6 min 22−38% B; 6−9 min 38−65% B; 9−12 min 65−80% B; 12−16 min 80−100% B; 16−19 min 100% B. The flow rate was constant at 0.25 mL min^−1^ for 12 min and was increased to 0.30 mL min^−1^ between 12 and 19 min. Each sample injection was followed by a wash injection, where the gradient was returned to its initial conditions in 14 min. The injection volume was 2 μL. The autosampler tray and column temperatures were set at 5 and 35 °C, respectively. The mass spectrometer ion source was operated with a probe capillary voltage of 2.5 kV and a sampling cone voltage of 30 V. The source and desolvation gas temperatures were set to 120 and 300 °C, respectively. The nitrogen gas desolvation flow rate was 600 L h^−1^, and the cone desolvation flow rate was 10 L h^−1^. The mass spectrometer was calibrated before the batch analysis across the range of *m*/*z* 50−1200 using a 0.5 mmol L^−1^ sodium formate solution prepared in isopropanol/water (90:10 *v*/*v*). Data were drift corrected during acquisition using a leucine enkephalin (*m*/*z* 556.2771) reference spray infused at 5 μL min^−1^, every 60 s. The scan time was set to 0.5 s and data were acquired in MS^E^ continuum mode in the range of *m*/*z* 50−1200. Six process replicates were acquired for candidate reference material plasma samples. Data acquisition and processing were carried out using MassLynx version 4.1 (Waters Corp., Milford, MA, USA). The mass spectrometry data have been deposited to the MetaboLights public repository [38] with the data set identifier MTBLS1919. Spectral features (Rt, *m*/*z* pairs) were extracted from UPLC–QTOF–MS data using Progenesis QI version 2.1 (Nonlinear Dynamics, Waters Corp., Milford, MA, USA). The procedure included retention time alignment, peak picking, deisotoping, integration, and grouping together adducts derived from the same compound.

### 4.5. Raw Data Conversion

Raw MS data in Waters RAW format where converted to mzML format in centroid mode using ProteoWizard’s msconvert software version 3.0 [39] with the following parameters: min distance = 0.01, snr = 1.

### 4.6. Python Package

TidyMS was implemented in Python 3.6. [40], and used the Pandas library to work with a data matrix in tabular form. Raw MS data were read using the pyopenms library, and routines for analyzing data were implemented using the Numpy [41], Scipy [42], and scikit-learn [27] libraries. Examples of web applications were built using the Bokeh library [29]. Source code, documentation, installation instructions, and Jupyter notebooks with Applications 1 and 2 are available at Github [33,34].

The pyopenms [30] package was used to read raw MS data in the mzML format. Peak picking functions were implemented using a CWT-based algorithm [35] with parameters optimized for chromatographic or spectral data. Additional functions were implemented for chromatogram building, for accumulating consecutive MS scans, detecting features using the centWave algorithm [36], and for feature detection. MSData, Chromatogram, Roi, and MSSpectrum were created using an object-oriented programming approach to abstract the aforementioned functionality and work with raw data. Appendix A (Appendix A) shows a description of the interaction between these objects.

Methods for filtering features in the data matrix are encapsulated in the DataContainer, Processor, and Pipeline objects. The DataContainer uses Pandas DataFrames to store the data matrix and the descriptors associated with features and samples. The sample mapping attribute of the DataContainer object associates sample types to sample classes, and it is used to set a default behavior for the different filters applied to the data matrix. The Pipeline object stores a list of Processor objects to build customized data curation strategies. Appendix A (Appendix A) shows a description of the interaction between the different objects used for data curation.

### 4.7. Statistical Analysis

Principal component analysis (PCA) models were built using the scikit-learn implementation. PCA was used in the pipeline to track data quality, identify potential outliers in the data set, identify sample clusters, and evaluate time-related systematic variance or signal drift. The LOESS smoother used in Batch Correction was implemented using the statsmodels [43] package.

## 5. Disclaimer

Certain commercial equipment, instruments, or materials are identified in this paper to adequately specify the experimental procedures. Such identification does not imply recommendation or endorsement by the National Institute of Standards and Technology; nor does it imply that the materials or equipment identified are necessarily the best for the purpose. Furthermore, the content is solely the responsibility of the authors and does not necessarily represent the official views of the National Institute of Standards and Technology.

## Figures and Tables

**Figure 1 metabolites-10-00416-f001:**
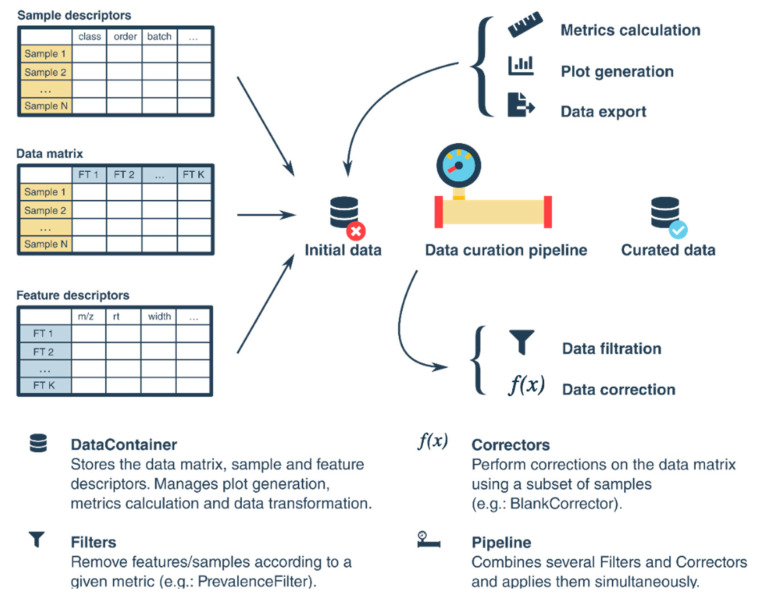
Graphical overview of the package workflow for data curation. Data are organized in a tabular form and processed using a combination of Filters and Correctors inside a Pipeline. In each processing stage, metrics and plots can be generated to track the data. After data curation, data can be exported in a variety of formats.

**Figure 2 metabolites-10-00416-f002:**
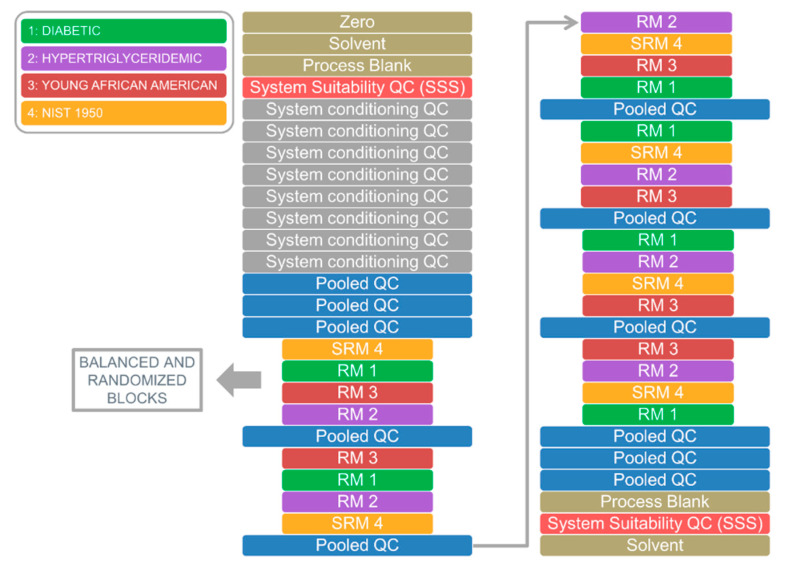
Run order of an untargeted metabolomics sample list. The experiment starts with a zero injection, followed by a reconstitution solvent injection, and a subsequent process blank injection. The system suitability QC sample (SSS) was run before column conditioning injections and after the analysis of the study and pooled/intrastudy quality control (QC) samples at the end of the experiment. Eight pooled/intrastudy QC sample injections addressed as “system conditioning QC” were used to equilibrate the analytical platform. The study samples corresponding to the National Institute of Standards and Technology (NIST) candidate RM 8231 and SRM 1950 were randomly analyzed within a template of pooled/intrastudy QC samples, where six blocks of four samples were injected, each block being individually balanced and randomized.

**Figure 3 metabolites-10-00416-f003:**
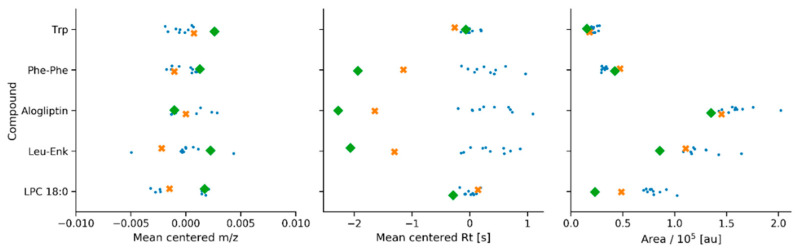
System suitability checks: reproducibility of *m/z*, Rt, and area values for five known chemical standards used in the system suitability QC sample (SSS). Values measured for ten injections of the SSS that were used to compute dispersion metrics for each analyte are shown with circle markers. Values obtained for an SSS before and after running the study samples are shown with a cross and a diamond marker, respectively.

**Figure 4 metabolites-10-00416-f004:**
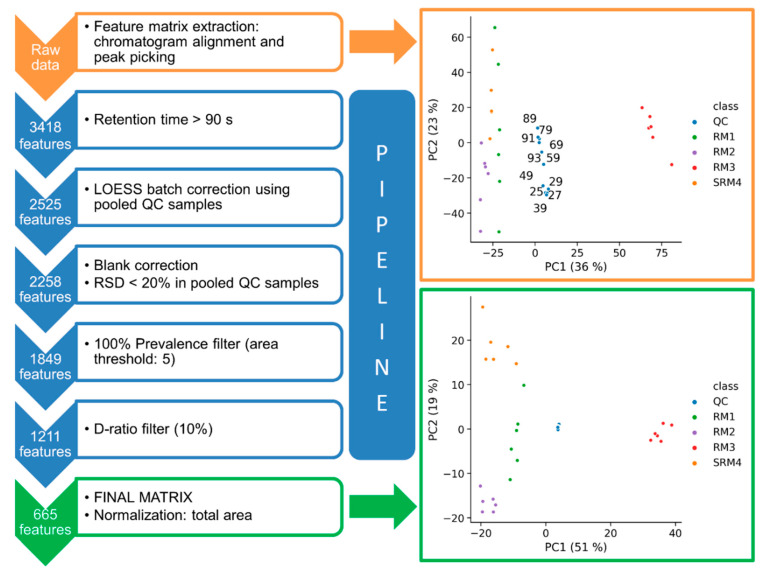
Processing scheme applied using the data curation Pipeline. The number of features conserved after each curation step are indicated on the left panel. Principal component analysis (PCA) score plots built before (orange) and after (green) processing the data. A better clustering of the pooled/intrastudy QC samples indicates that the data are more robust after curation.

**Table 1 metabolites-10-00416-t001:** Comparison of available free, open-source tools used in the preprocessing stage of liquid chromatography–mass spectrometry (LC–MS)-based metabolomics workflows.

	Tool	TidyMS	MZmine2	XCMS ^a^	MS-DIAL	nPYc- Toolbox	SECIM TOOLS	Notame
Features								
Language		Python	Java	R	C#	Python	Python	R
Raw data preprocessing Tools (e.g., feature detection and correspondence)		✅	✅	✅	✅	❌	❌	❌
Data curation		✅	✅	❌	✅	✅	✅	✅
QC-based batch correction		✅	❌	❌	❌	✅	❌	✅
Quality metrics (e.g., RSD in samples and references, PCA plots)		✅	✅	❌	✅	✅	✅	✅
Normalization, imputation, scaling		✅	✅	❌	✅	✅	✅	✅
Feature annotation		❌	✅	❌	✅	❌	❌	❌

^a^ Even if some features are marked as missing, there is a wide variety of libraries for the R language that use XCMS as a starting point to address them.

## Data Availability

Source code, documentation, and installation instructions are available online at Github: https://github.com/griquelme/tidyms.

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
