# Peer review of "A Python-Based Pipeline for Preprocessing LC–MS Data for Untargeted Metabolomics Workflows"

_metabolites, 2020, doi:10.3390/metabo10100416_

Round 1
Reviewer 1 Report
The authors present their new tool TidyMS which was developed to work as a pipeline for processing untargeted metabolomics data from LC-MS devices. Especially in the paper, they present two applications that can be performed with TidyMS.
In general, I am a big fan of open-source analysis tools from the mass spectrometry field especially when written in Python :-) On the first glance, the package sounds very promissing, the paper is well written, the methods and algorithms described in the paper that were utilized in the tool seem to be reasonably applied and one can see that you put much love into the tool. I was happy that the installation was straight forward.
However, to convince me in total, the paper / documentation needs some improvements and additional data. In the following, I will provide an unordered list with all my issues. I read your manuscript carefully. If I state an issue which was already addressed in the manuscript, may be it was described in a way that I missed it.
Main manuscript:
1.1) Since you are using the journal to advertise your tool in the paper, please put the repository / download link in more prominent places as, e.g., in the abstract. It the beginning, I didn't find the link to the github repository
1.2) Line 76: "... some of its capabilities, including ...": what are the remaining capabilities?
1.3) Line 157: Please provide the graphic as a vector graphic. It contains only pictograms and text, should be no problem to vectorize it.
1.4) I really would like to re-run your two appliation pipelines especially with my data. Please provide a folder in the git repository, say "example pipelines" with 2 - 3 python scripts that implement these pipelines. Of course, this should be mentioned in the documentation.
1.5) Line 209: Please add a legend in the figure.
1.6) Line 249-250: "In addition, metrics can be computed ...": Which metrics, can you name some of them or point in the paper at them?
1.7) I am missing some discussion about the false negative rate. How many of true metabolites slip through your filters when applying them? Did you do some experiments? Have you found all metabolites in your datasets or in the NIST dataset?
1.8) Please provide a small test data set containing all necessary files. Here, I must admit that after spending almost one day running your tool on my data (and I used a rather small dataset), I couldn't finish any analysis. It would be great to have next to a set of measurement files also a working python script of the complete quickstart documentation. When putting all code snippets together, the file doesn't work. For some functions, I need additional files like "sample_metadata.csv" in feature correspondence section where you roughly discribed, who it should look like. Please provide an example file. In the Section "Computing feature metics", some reference-materials is being loaded. Is this an additional file? How should it look like? I hope, you got my point. Please store this complete dataset including all measurements, python scripts and additional meta files in a repository as MetoLights.
1.9) Please put also the data for your both applications as separate repositories into MetaboLights
1.10) Can you state how it could be possible to add a metabolite component into your pipeline or isn't it desired in your system?
1.11) I guess, your unique feature among the tools MZmine2, XCMS, MSDial is your clever QC application. Please setup a (binary) table tools vs. features to show which tool contains which feature to underline your uniqueness.
Tool
2.1) Is the tool capable of parallel computing? During the feature detection, only one core is being used. Since all files are processed sequencially, it should be straight forward to parallelize the function tidyms.detect_features(). For instance, you could add an additional parameter "num_cpu" which by detault uses all available processors unless defined otherwise by the user.
2.2) How do I store my results? Somehow I missed it completely in the documentation. Are you using standard formats as mzTab-m (by Hoffmann et al.)? If this is not the case, I really encourage you to do so.
Supplementary Information
3.1) Please provide a table of contents
3.2) Section "Initial feature clustering": Haha, you guys are funny. In one sentence you are claiming that DBSCAN is a non-parametric algorithm, whereas in the following sentence explain its first parameter epslion ;-) Actually, DBSCAN requires two parameters as you state correctly. So I don't understand why you claim it is non-parametric. Please explain it or leave this statement out.
3.3) Section "Cluster assessment": You are using a GMM to deconvolute subclusters in your initial clusters. Are you giving the features a weight (as for instance the abundance of the feature) or are all features equally weighted?
Reviewer 2 Report
The present study describes a workflow for data evaluation from untargeted metabolomics experiments. The focus is on excluding unsuitable features from a data matrix.The paper is well written, but the approach is not really new and for the most part already established in common software. It would certainly help to better emphasize the unique advantages of the proposed workflow. In lines 102-103, the authors briefly mention that there are other comparable software alternatives. At this point, it would be helpful to highlight the strengths of the study more precisely.
In line 175 it could help to explain some criteria for quality assurance in non-targeted laboratories.
Reviewer 3 Report
The authors introduce TidyMS, a Python package covering the whole data preprocessing and pretreatment in untargeted metabolomics, from peak picking and alignment, to data correction and feature filtering. This is one of the areas of research of metabolomics where tools enabling the use of fast, reproducible and exchangeable workflows is most needed.
The article is well written, clear and scientifically sound. The package is well designed and its applicability to real-world cases is demonstrated through two examples.
I am only missing mentioning other existing tools with similar possibilities such as WorkFlow4Metabolomics. A critical comparison between both of them in terms of features, speed, reproducibility and exchangeability would be much appreciated.
In addition, some recent reviews on untargeted workflows for metabolomics should be cited in the introduction: http://doi.org/10.3390/metabo9120308, http://doi.org/10.3390/metabo10040135, http://doi.org/10.1016/j.aca.2019.12.062
